# Scalp dose analysis for transient and permanent alopecia following conventional cranial irradiation using Image Guided Radiotherapy (IGRT): A prospective study

Bongkot Jia-Mahasap[1☺], Wannapha Nobnop[1‡], Patumrat Sripan[2☺], Ekkasit Tharavichitkul[1‡], Somvilai Chakrabandhu[1‡], Pitchayaponne Klunklin[1‡], Wimrak Onchan[1‡], Pooriwat Muangwong[1‡], Imjai Chitapanarux[1]*

**1** Division of Radiation Oncology, Department of Radiology, Faculty of Medicine, Chiang Mai University, Chiang Mai, Thailand, **2** Research Institute for Health Sciences, Chiang Mai University, Chiang Mai, Thailand

☺ These authors contributed equally to this work.
‡ WN, ET, SC, PK, WO and PM also contributed equally to this work.
* imjai@hotmail.com

## Abstract

### Purpose

The dosimetry of scalp dose was prospectively studied and correlated with alopecia following conventional cranial irradiation in primary brain tumors patients.

### Materials and methods

Patients with primary brain tumors who required conventional radiotherapy were enrolled. A hairline marker was applied to the patient's scalp to identify the entire scalp region. The maximal dose to 2% volume of interest (D2) for the entire scalp region were obtained. The radiation dosages at the localized hair-loss areas were evaluated during the final week of RT (transient alopecia) and six months after completing RT (permanent alopecia). Kruskal-Wallis tests were used to compare the dosimetric parameter values with statistical significance set as $p < 0.05$.

### Results

Forty-eight patients were included in the analysis. The prescribed radiation doses ranged from 50.4 to 60.0 Gy. Thirty-two patients experienced alopecia (27 transient and 5 permanent). The median D2 values adjusted for the entire scalp were higher in the alopecia group (38.40 Gy for transient alopecia and 47.84 Gy for permanent alopecia vs 11.90 Gy for no alopecia, $p < 0.001$). The D2 value was determined as a predictive parameter for alopecia. The threshold values for transient and permanent alopecia over the entire scalp were 22.15 Gy and 36.81 Gy, respectively. At the localized hair-loss areas, the D2 values for transient and permanent alopecia were higher at 44.82 Gy and 50.00 Gy, respectively. The radiation intensity at the localized hair-loss areas was also related to the severity of alopecia, with D2

**Data Availability Statement:** All relevant data are within the manuscript and its Supporting Information files.

**Funding:** The author(s) received no specific funding for this work.

values of 35.14 Gy and 46.39 Gy for clinically assigned grade 1 and grade 2 transient alopecia, respectively, with the D2 value being even higher for permanent alopecia.

## Conclusions

The D2 parameter value could be used to predict the type and severity of alopecia.

## Introduction

Radiotherapy (RT) is an important treatment for primary brain tumors. Typically, external-beam RT is utilized for intracranial lesions during which the radiation beam generated by a linear accelerator machine directly penetrates the cranium into the brain parenchyma to eradicate tumors and delivers a radiation dose from 45 to 60 Gy depending on the tumor subtype [1, 2]. Localized alopecia along the radiation path is a typical side effect of RT. Transient alopecia occurs approximately 2 to 3 weeks after commencing RT and usually resolves within 3–6 months [3]. RT-induced apoptosis of hair bulges containing stem cells can result in permanent alopecia, which can have deleterious psychological effects on quality of life and self-esteem [4].

Previous research has demonstrated a clinical correlation between alopecia and whole brain RT for the treatment of brain metastatic [5–8]. They all used a hypo-fractionated radiation dose regimen (ranging from 2.5 Gy to 4 Gy per fraction [Fx]) which typically delivers a greater absorbed radiation dose to healthy tissue. There were few investigations examining the relationship between focalized RT and hair loss in patient with primary brain tumors receiving 1.8–2.0 Gy per Fx [3, 9]. A notable study conducted by Lawenda and colleagues [9] found that hair follicles exposed to 43 Gy of radiation had a 50% chance of developing permanent alopecia [9]. However, they estimated hair follicle dosage by calculating the radiation dosage delivered by each radiation beam passing through the scalp region of interest using extremely complex calculations, which could be difficult to implement in practice. In other studies, hair follicles were identified by contouring tissue between skin and scalp at skin depths ranging from 3 to 7 mm after hypo-fractionated whole-brain RT [3, 5–8, 10].

The human hair follicle is traditionally divided into three sections: the infundibulum, the isthmus, and the inferior portion. The bulge region, which is located in the isthmus and inferior portion, is the most important subunit in which stem cells are pooled [11]. The length of the hair follicle from the epidermis is between 3.86 mm and 4.16 mm [11, 12]. Exploiting this crucial data, we decided to study the area of interest on the scalp by using automated tissue contouring to distinguish hair follicles from skin at a depth of 4 mm.

We hypothesize that permanent alopecia resulting from irreversible damage to the hair follicle after focal RT for primary brain tumors is directly proportional to the hair follicle radiation dose. A reduction in the scalp dose without compromising target coverage may reduce hair loss and thereby improve the quality of life of the patients. Therefore, the purpose of this prospective study was to determine the scalp dose that is associated with alopecia and to identify additional variables (such as age and gender) that may have an influence.

## Materials and methods

This prospective study enrolled patients who met the following criteria: 1) Age 18 to 70 years, 2) Eastern Cooperative Oncology Group (ECOG) performance status 0–2 [13], 3) Primary intracranial solid tumors as determined by histopathology or classical imaging for benign brain tumors, 4) Radiation delivery using conventional fractionation scheme, and 5)

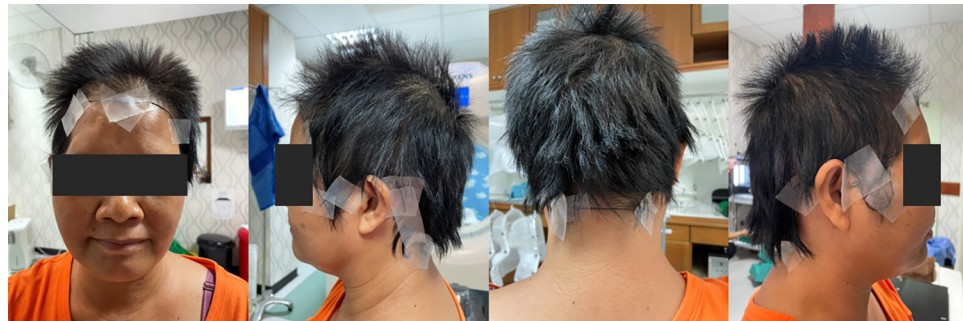

**Fig 1. A hairline marker on a scalp before CT simulation.**

Concurrent temozolomide chemotherapy was allowed, which has a low rate of chemotherapy-induced transient alopecia [14]. The exclusion criteria were patients 1) who had previously undergone cranial irradiation, 2) who were already showing signs of baldness, 3) who had received a radiation dose greater than 2.5 Gy/Fx, and/or 4) who had previously undergone chemotherapy. Ethical approval was granted by the Research Ethics Committee, Faculty of Medicine, Chiang Mai University (Grant no. 236/2019). Written consent was obtained from each of the participants.

Computed tomography (CT) simulation with 3-mm thick slices was conducted on the eligible patients. During this procedure, a thermoplastic mask with a hairline marker at the scalp was applied (Fig 1), which assisted in delineating the scalp region. After the simulation, the CT images were registered in an Oncentra master plan contouring system (Elekta, Sweden). Subsequently, the target volumes (gross tumor volume, GTV; clinical target volume, CTV; planned treatment volume, PTV) and critical structures (organs at risk, OARs) were defined based on the primary tumor type. The volume of the entire scalp was measured based on the hairline marker. Finally, an automated ring region with a maximum thickness of 4 mm was generated between the skin and the cranium at this location (Fig 2).

Radiation had been administered to the patients using image-guided RT (IGRT), which included volumetric arc therapy (VMAT) and helical tomotherapy (HT). Target dosages were evaluated in accordance with the International Commission on Radiation Units and Measurements (ICRU) 83 guidelines [15], which stipulate that the prescription radiation dose coverage must be over at least 50% of the PTV (D50). The maximum dosage applied to a 2% volume of the PTV (D2) needed to be less than 107% of the prescribed dose while the minimum dosage to a 98% volume of the PTV (D98) needed to be greater than 95% of the prescribed dose. The Quantitative Analysis of Normal Tissue Effects in the Clinic (QUANTEC) guidelines were employed to assess the irradiation of OARs [16]. For the scalp volume, we defined radiation dosage constraints without compromising the target coverage at three different doses: the primary constraint was D2 ≤ 20 Gy, followed by ≤ 30 Gy and ≤ 40 Gy as secondary and tertiary constraints, respectively. The mean (Dmean), median (D50), and D2 radiation doses to the scalp region, as well as the absolute scalp volumes receiving doses of 20 Gy (V20) and 40 Gy (V40) were obtained and recorded.

## Alopecia assessment

Right lateral, left lateral, vertex, and occipital regions of the scalp were objectively photographed at five distinct time intervals: before beginning RT, during the final week of RT, and three, six, and 12 months after completing RT. Later, three independent investigators assigned alopecia grades to each photo. The Common Terminology Criteria for Adverse Events

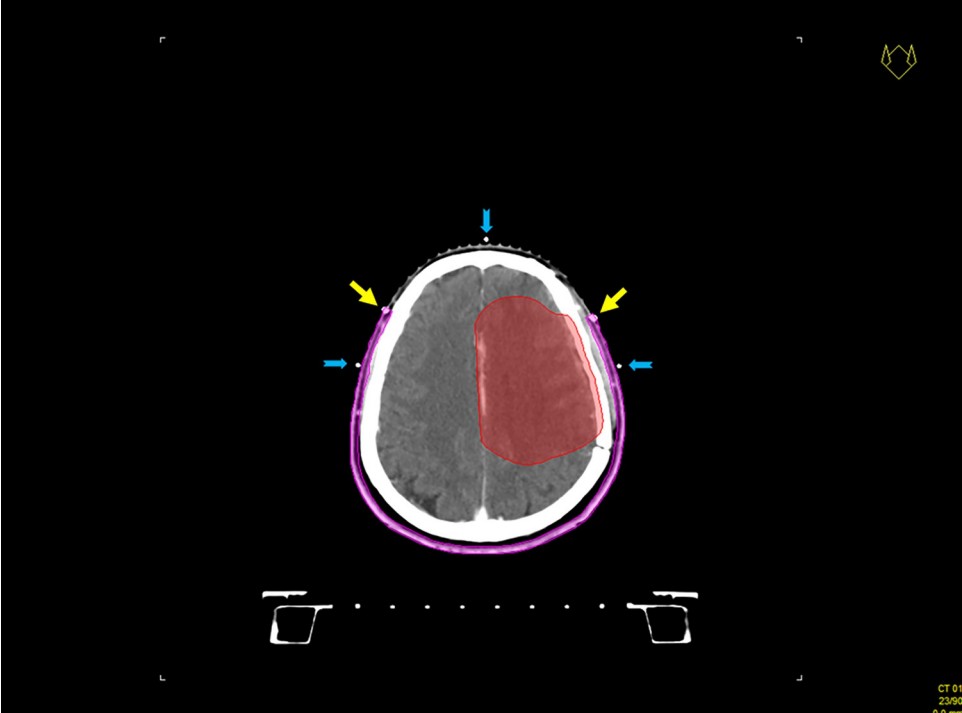

**Fig 2. The scalp volume (purple).** The yellow arrows point toward the pre-applied hairline marker. The blue arrows indicate the isocenter.

(CTCAE) version 5.0 was used to assess the severity of alopecia (grade 1, hair loss < 50% of normal; grade 2, hair loss ≥ 50% of normal) [17]. The areas of localized hair loss that occurred during the final week of RT (designated as transient alopecia) and six months after completing RT (designated as permanent alopecia) had their radiation dosages re-evaluated by wiring a scalp marker to a thermoplastic mask, CT simulation of this wired mask, and assigning the images to their previous counterparts used for radiation planning by applying the isocenter as the matching point (Fig 3). RT research (CERR) software was used to analyze the dosimetric parameters for the region of interest using Dicom RTdose data from the prior treatment planning system that delivered radiation to the patient [18]. The radiation dosimetry was recorded as the Dmean, D50, D2, V20, and V40 values.

## Sample size

This study sample size was determined according to Harris's recommendation [19]; for 5 or fewer predictors, the number of subjects should exceed the number of independent variables by 50 (n > 50+m). We assessed the relationship between alopecia and three predictors: radiation dose, age, and gender. Consequently, the sample size needed to be 74 patients.

## Statistical analysis

Descriptive analyses were summarized as medians with interquartile range (IQR) for continuous variables and as frequencies and proportions for categorical variables. The values of the dosimetric parameters (Dmean, D50, D2, V20, and V40.) were summarized using the medians and IQR. Kruskal-Wallis tests were performed to compare the medians for the dosimetric parameters for the entire scalp of patients without alopecia, or with transient (including

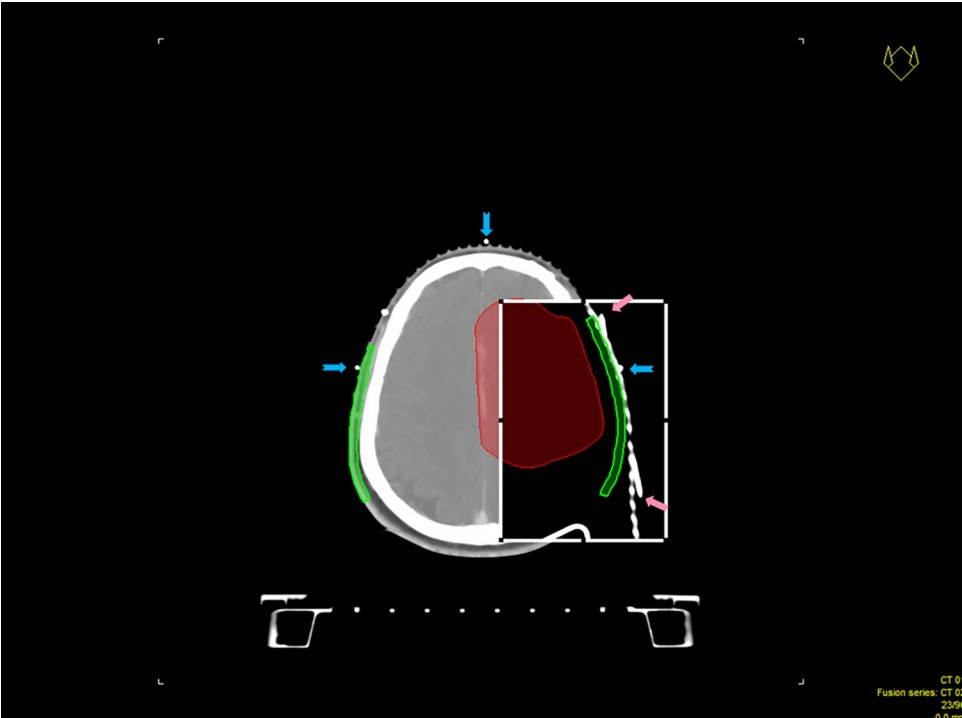

**Fig 3. The CT simulation of a wired mask of the hair-loss region assigned by using previous CT images.** The blue arrows indicate the isocenter as the matching point. The pink arrows elucidate the correlated hairline marker for localized alopecia. The specific scalp region (green) could then be identified.

different grades) or permanent alopecia. Pair-wise comparisons were conducted using Dunn's Kruskal-Wallis Multiple Comparison tests. Wilcoxon rank-sum tests were used to compare the dosimetric parameter values for the localized hair-loss areas of patients with grade I or II transient alopecia and between those with and without alopecia. Associations between transient alopecia grade and permanent alopecia were analyzed using Fisher's Exact tests. After plotting receiver operating characteristic (ROC) curves, relevant results including the area under the curve (AUC) with sensitivity and specificity were estimated to determine the diagnostic values of the dosimetry parameters. All statistical analyses were performed using Stata version 16 (StataCorp LP, College Station, TX, USA). Statistical significance for two-tailed tests was assessed using an alpha value of 0.05.

## Result

Fifty-six patients satisfied the eligibility requirements. Among them, 8 did not undergo CT simulation, leaving 48 patients included in the analysis. Meningioma was the most prevalent tumor type (21 patients, 44%), followed by glioma (17 patients, 35%) and other types (10 patients, 21%). The median age was 51 (47–56 years) and the majority were female (37 patients, 77%). The median follow-up was 27 months (14–33 months). The prescribed RT doses ranged from 50.4 to 60.0 Gy (1.8–2.0 Gy/Fx). Thirty-two out of 48 patients (66.7%) experienced hair loss, with 27 experiencing transient alopecia and 5 experiencing permanent alopecia. Age, gender, RT technique, and scalp volume parameters were unrelated to hair loss, whereas the PTV volume and radiation intensities on the scalp region were statistically significantly associated with alopecia, as presented in Table 1.

**Table 1. Characteristics of the participants.**

| Characteristics | No alopecia | Transient alopecia | Permanent alopecia | P-value |
|---|---|---|---|---|
| | N = 16 | N = 27 | N = 5 | |
| Median age (IQR), year | 51 (47–56) | 52 (44–60) | 56 (46–58) | 0.890[a] |
| Gender, N (%) | | | | 0.171[b] |
| Male | 2 (18.18) | 9 (81.82) | 0 (0.00) | |
| Female | 14 (37.84) | 18 (48.65) | 5 (13.51) | |
| RT technique, N (%) | | | | 0.696[b] |
| VMAT | 9 (39.13) | 12 (52.17) | 2 (8.70) | |
| HT | 7 (28.00) | 15 (60.00) | 3 (12.00) | |
| Median PTV volume (IQR), cm³ | 34 | 207.89 | 164.63 | <0.001[a] |
| | (13.76–40.43) | (97.31–355.43) | (116.92–327.93) | |
| Median scalp volume (IQR), cm³ | 226.08 | 224.62 | 201.97 | 0.531[a] |
| | (209.98–250.36) | (203.10–237.96) | (190.37–236.87) | |
| Median Dmean for the entire scalp (IQR), Gy | 2.28 | 9.70 | 9.55 | <0.001[a] |
| | (1.67–3.42) | (6.72–16.08) | (9.43–15.10) | |
| Median D50 for the entire scalp (IQR), Gy | 0.42 | 7.61 | 6.81 | <0.001[a] |
| | (0.31–0.73) | (2.43–14.91) | (5.71–10.57) | |
| Median D2 for the entire scalp (IQR), Gy | 11.90 | 38.4 | 47.84 | <0.001[a] |
| | (10.68–15.80) | (28.24–43.11) | (36.81–48.21) | |
| Median V20 for the entire scalp (IQR), cc | 0.00 | 37.75 | 30.25 | <0.001[a] |
| | (0.00–0.12) | (14.40–73.95) | (28.48–55.62) | |
| Median V40 for the entire scalp (IQR), cc | 0.00 | 3.78 | 12.25 | <0.001[a] |
| | (0.00–0.00) | (0.28–7.84) | (3.63–17.83) | |

[a]Determined using a Kruskal-Wallis test

[b]determined using a Fisher's Exact test.

### Radiation dosimetry and clinical alopecia correlation

The Dmean value for the entire scalp was substantially higher in the alopecia groups (9.70 Gy for transient alopecia and 9.55 Gy for permanent alopecia vs. 2.28 Gy for without alopecia, $p < 0.001$), as was the D50 value (7.61 Gy and 6.81 Gy vs. 0.42 Gy, respectively, $p < 0.001$), and the D2 value (38.40 Gy and 47.84 Gy vs. 11.90 Gy, respectively, $p < 0.001$). Furthermore, the V20 and V40 values were considerably higher in the alopecia group, as demonstrated in Table 1. The D2 and V20 values were also determined as predictive parameters of alopecia. The threshold values for transient and permanent alopecia were 22.15 Gy and 36.81 Gy for D2, and 6.63 cc and 16.36 cm³ for V20, respectively, as shown in Table 2. When measured at the site of localized hair loss, one patient who experienced transient alopecia did not undergo a re-evaluation of the localized hair-loss areas dosage. Therefore, the analyses were conducted on the scalp doses of 26 patients with transient alopecia, as shown in Table 3. The permanent alopecia group had greater values for all five dosimetric parameters. The median Dmean, D50, D2, V20, and V40 values were 31.07 Gy, 30.85 Gy, 50.00 Gy, 28.08 cm³, and 12.26 cm³, respectively.

### Impact of radiation dose on the severity of alopecia

Grade 1 alopecia was diagnosed in 3 of the 32 patients who experienced hair loss following the completion of RT, while the remaining 29 patients were diagnosed with grade 2 alopecia. Medians of the dosimetry parameter values were used to determine any correlations between

**Table 2. Median radiation dosimetry parameter values for the entire scalp volume.**

| Parameters | Sensitivity (%) | Specificity (%) | AUC | 95% CI (AUC) | Cutoff value | Youden's index |
|---|---|---|---|---|---|---|
| **Transient Alopecia** | | | | | | |
| Mean Dose | 0.938 | 0.938 | 0.982 | 0.956–1.000 | 4.17 Gy | 0.875 |
| D50 | 0.844 | 1.000 | 0.963 | 0.918–1.000 | 2.41 Gy | 0.844 |
| D2 | 0.969 | 1.000 | 0.994 | 0.981–1.000 | 22.15 Gy | 0.969 |
| V20 | 0.969 | 1.000 | 0.994 | 0.981–1.000 | 6.63 cc | 0.969 |
| V40 | 0.875 | 0.938 | 0.922 | 0.854–0.990 | 0.18 cc | 0.812 |
| **Permanent Alopecia** | | | | | | |
| Mean Dose | 1.000 | 0.553 | 0.823 | 0.695–0.952 | 6.33 Gy | 0.553 |
| D50 | 1.000 | 0.632 | 0.817 | 0.696–0.938 | 3.09 Gy | 0.632 |
| D2 | 0.800 | 0.737 | 0.813 | 0.685–0.942 | 36.81 Gy | 0.537 |
| V20 | 1.000 | 0.605 | 0.810 | 0.682–0.939 | 16.36 cc | 0.605 |
| V40 | 0.900 | 0.658 | 0.776 | 0.616–0.937 | 1.62 cc | 0.558 |

them and the severity of alopecia. The Dmean values for the entire scalp were substantially less in the group without alopecia than in the group with alopecia: 2.28 Gy vs. 9.02 Gy and 9.70 Gy for no alopecia vs. grades 1 and 2, respectively ($p < 0.001$) as were the D50 and D2 values (0.42 Gy vs. 5.00 Gy and 7.61 Gy, and 11.90 Gy vs. 34.15 and 38.52 Gy, respectively, $p < 0.001$), as presented in Table 4. Furthermore, the V20 and V40 values of the non-alopecia group were both 0.00 cm$^3$, both of which were significantly greater for alopecia grades 1 and 2. The focal scalp dosage of radiation intensity at the localized hair-loss area was not re-evaluated for 1 patient with grade 1 alopecia but was for 2 cases of grade 1 alopecia and 29 cases of grade 2 alopecia: none of the dosimetric parameter values were statistically significantly different between the two grades (Dmean: 20.40 Gy vs. 25.75 Gy, $p = 0.31$; D50: 20.67 Gy vs. 26.45 Gy, $p = 0.24$; D2: 35.14 Gy vs. 46.39 Gy, $p = 0.35$; V20: 56.16 cm$^3$ vs. 28.08 cm$^3$, $p = 0.748$, and V40: 2.59 cm$^3$ vs. 4.22 cm$^3$, $p = 0.421$).

## Discussion

Alopecia is a common side effect following conventional RT, as was also seen in 66.7% of patients in the present prospective study. Significant factors correlating with clinical alopecia included the radiation intensity in the scalp region. The median Dmean values for the entire scalp region were 9.70 and 9.55 Gy in the groups with transient and permanent alopecia, respectively, compared to 2.28 Gy in the group without alopecia. This is in concert with the results of a clinical investigation conducted in Italy [20], the authors of which reported Dmean value of 11.6 Gy and 14.1 Gy for the entire scalp of patients who developed transient or permanent alopecia after undergoing RT, respectively, compared to 3.1 Gy for patients without alopecia. Differences in race and the depth of hair follicle bulge definition (5 mm) might explain the higher radiation dose in the scalps of persons with RT-induced alopecia. Moreover,

**Table 3. Median radiation dosimetry parameter values for the localized hair-loss areas.**

| Parameters | Transient Alopecia (N = 26) | Permanent Alopecia (N = 5) |
|---|---|---|
| Median Dmean (IQR), Gy | 24.70 (19.96–28.79) | 31.07 (27.80–41.92) |
| Median D50, (IQR), Gy | 25.28 (18.27–28.91) | 30.85 (28.92–43.09) |
| Median D2, (IQR), Gy | 44.82 (39.23–48.65) | 50.00 (48.94–51.93) |
| Median V20 (IQR), cm$^3$ | 27.78 (9.68–63.25) | 28.08 (26.88–51.68) |
| Median V40 (IQR), cm$^3$ | 3.68 (0.28–7.85) | 12.26 (3.08–13.14) |

**Table 4. Correlations between median radiation dosimetry parameter values and the severity of alopecia.**

| Parameters | Severity of Alopecia | | | P-value |
|---|---|---|---|---|
| | **No Alopecia** | **Grade 1** | **Grade 2** | |
| | **N = 16** | **N = 3** | **N = 29** | |
| **The entire scalp** | | | | |
| Median Dmean (IQR), Gy | 2.28 | 9.02 | 9.70 | <0.001[a] |
| | (1.67–3.42) | (7.80–18.70) | (6.72–15.96) | |
| Median D50 (IQR), Gy | 0.42 | 5.00 | 7.61 | <0.001[a] |
| | (0.31–0.73) | (3.09–20.20) | (2.91–14.34) | |
| Median D2 (IQR), Gy | 11.90 | 34.15 | 38.52 | <0.001[a] |
| | (10.68–15.80) | (24.96–41.54) | (34.81–45.51) | |
| Median V20 (IQR), cm$^3$ | 0.00 | 41.36 | 32.05 | <0.001[a] |
| | (0.00–0.12) | (18.81–107.88) | (16.36–72.04) | |
| Median V40 (IQR), cm$^3$ | 0.00 | 0.28 | 4.27 | <0.001[a] |
| | (0.00–0.00) | (0.00–5.18) | (1.36–11.76) | |
| **Localized hair-loss areas** | | Grade 1 | Grade 2 | |
| | | N = 2 | N = 29 | |
| Median Dmean (IQR), Gy | | 20.40 | 25.75 | 0.310[b] |
| | | (15.89–24.92) | (20.21–29.14) | |
| Median D50 (IQR), Gy | | 20.67 | 26.45 | 0.241[b] |
| | | (16.27–25.07) | (19.43–29.29) | |
| Median D2 (IQR), Gy | | 35.14 | 46.39 | 0.348[b] |
| | | (24.67–45.61) | (40.75–49.57) | |
| Median V20 (IQR), cm$^3$ | | 56.16 | 28.08 | 0.748[b] |
| | | (6.29–106.02) | (12.03–61.95) | |
| Median V40 (IQR), cm$^3$ | | 2.59 | 4.22 | 0.421[b] |
| | | (0.00–5.18) | (1.37–11.4) | |

[a]Obtained using a Kruskal-Wallis test

[b]obtained using a Wilcoxon rank-sum test.

V20 > 5.2 cm$^3$ for transient alopecia and V40 $\geq$ 2.2 cm$^3$ for permanent alopecia were the strongest predictors in their study [20]. In the present study, the D2 and V20 values were determined to be potentially predictive parameters for alopecia. The threshold values for transient and permanent alopecia were 22.15 and 36.81 Gy for D2, and 6.63 and 16.36 cm$^3$ for V20, respectively. However, regarding the correlation between radiation dose and the severity of alopecia, we found that a higher D2 value was associated with a greater severity grade of alopecia, whereas V20 was lower in the grade 2 alopecia group. This outcome could have been influenced by the limited number of patients with grade 1 alopecia, so we recommend conducting a future study with a greater number of patients to more accurately assess this correlation. Another European research group reported that an absolute radiation dose of 12.6 Gy to the hair follicle depth (4 mm) was associated with transient alopecia after conventional fractionation [21]. They measured visible alopecia in two directions using a ruler and applied it to a standardized picture of the human scalp. The absolute radiation dosage was automatically established at the site of alopecia by the specific treatment planning system. The potential discrepancy between the image and the actual region of hair loss might have restricted the data analysis in their report. Other factors, including chemotherapy, medication use, and mental stress levels are not predictive of radiation-induced alopecia [9, 21]. Lawenda et al. [9] applied an extremely complex formula to establish the radiation dosimetry associated with clinical

focal hair loss in their study. The D50 value for hair follicle dose was 43 Gy, which was found to be correlated with 50% of patients who developed permanent alopecia. In contrast, we found that patients who developed permanent alopecia had a median D50 value for localized hair loss of 30.85 Gy. We also found correlations between the Dmean and D50 values for the localized hair-loss areas and the severity of alopecia: the Dmean values for grades 1 and 2 were 20.40 Gy and 25.75 Gy while the D50 values were 20.67 Gy and 26.45 Gy, respectively. Lawenda et al. [9] also found a correlation between higher radiation dosimetry values and alopecia grade severity: the Dmean values for grade 1 and grade 2 alopecia were 31.6 and 46.1 Gy while the D50 values were 32.9 and 48 Gy, respectively. In another study conducted in the US [22], the D50 values for grades 1 and 2 alopecia were 31.9 and 41.5 Gy, respectively, thereby suggesting that this is a significant parameter for predicting radiation-induced alopecia. Again, differences in ethnicity may explain the disparity in radiation intensity causing clinical alopecia. Additionally, we found that larger treatment volumes were also associated with clinical alopecia due to their potential to attach the scalp region.

The definition of permanent alopecia is a controversial topic. Persistent radiation-induced alopecia was defined by Phillips and colleagues [22] as incomplete hair regrowth six months after the completion of RT whereas Lawenda and colleagues [9] defined it as 12 months. We used the former definition in our investigation. We also reassessed the radiation dose at the permanent hair-loss region of the patients with permanent alopecia. Nevertheless, we also photographed the scalps of the patients with permanent alopecia for 12 months following the completion of RT and found that those with permanent alopecia at 6 months also had it after 12 months.

In our cancer-treatment center, IGRT is the preferred radiation therapy modality for patients receiving treatment for cranial tumors. Thus, all of the patients included in the present study had undergone IGRT. This technique enhances precision by ensuring that radiation is delivered accurately to the targeted area by accounting for any patient movements or anatomical changes. By targeting radiation more precisely, IGRT also helps minimize unnecessary radiation exposure to the scalp. Although IGRT did not show a significant improvement in tumor control or reduction of radiation-induced toxicity for intracranial RT, further research is needed to evaluate the potential benefits of IGRT compared to conventional IMRT.

This prospective study's strength was the detailed evaluation of alopecia. We applied a hairline marker to identify the scalp region as well as the regions of alopecia. We also evaluated photographs of the scalp to assist researchers in better grading alopecia. In addition, we provided several dosimetric parameters for potentially predicting radiation-induced alopecia that are easy to calculate in clinical practice. Of them, a higher D2 value was correlated with localized hair loss. The median D2 values for transient and permanent alopecia were 38.40 and 47.84 Gy, respectively for the entire scalp but higher in the localized hair-loss areas (44.82 and 50.00 Gy, respectively). A higher D2 value was also detected in the higher alopecia grade (35.14 and 46.39 Gy for grades 1 and 2, respectively). Limitations of our analysis are not using a quality-of-life questionnaire assessment and not using advanced modeling to predict alopecia following conventional photon RT. For future research, a larger sample size would help to confirm which parameters are useful for predicting radiation-induced alopecia.

## Conclusion

We discovered that all five dosimetric parameters investigated (Dmean, D50, D2, V20, and V40) were associated with radiation-induced alopecia. Among them, D2 proved to be the best for predicting the propensity and grade of alopecia post-RT: the greater the D2 value, the higher the alopecia grade.

## Supporting information

**S1 File. Contained the characteristics of each patient.**
(XLSX)

## Author Contributions

**Conceptualization:** Bongkot Jia-Mahasap, Wannapha Nobnop, Imjai Chitapanarux.

**Formal analysis:** Bongkot Jia-Mahasap, Patumrat Sripan.

**Investigation:** Bongkot Jia-Mahasap, Ekkasit Tharavichitkul, Somvilai Chakrabandhu, Pitchayaponne Klunklin, Wimrak Onchan, Pooriwat Muangwong.

**Methodology:** Bongkot Jia-Mahasap, Wannapha Nobnop, Patumrat Sripan.

**Supervision:** Imjai Chitapanarux.

**Writing – original draft:** Bongkot Jia-Mahasap, Wannapha Nobnop, Patumrat Sripan, Ekkasit Tharavichitkul, Somvilai Chakrabandhu, Pitchayaponne Klunklin, Wimrak Onchan, Pooriwat Muangwong, Imjai Chitapanarux.

**Writing – review & editing:** Bongkot Jia-Mahasap, Wannapha Nobnop, Patumrat Sripan, Ekkasit Tharavichitkul, Somvilai Chakrabandhu, Pitchayaponne Klunklin, Wimrak Onchan, Pooriwat Muangwong, Imjai Chitapanarux.

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
