## [Decision Letter · Decision Letter 0]

10 Jul 2024

PONE-D-23-41670Scalp dose analysis for transient and permanent alopecia following conventional cranial irradiation using Image Guided Radiotherapy (IGRT): A prospective studyPLOS ONE

Dear Dr. Chitapanarux,

Thank you for submitting your manuscript to PLOS ONE. After careful consideration, we feel that it has merit but does not fully meet PLOS ONE’s publication criteria as it currently stands. Therefore, we invite you to submit a revised version of the manuscript that addresses the points raised during the review process.

We look forward to receiving your revised manuscript.

Kind regards,

Li-Ping Liu

Academic Editor

PLOS ONE

Reviewers' comments:

Reviewer's Responses to Questions

**Comments to the Author**

1. Is the manuscript technically sound, and do the data support the conclusions?

Reviewer #1: Yes

2. Has the statistical analysis been performed appropriately and rigorously? 

Reviewer #1: Yes

3. Have the authors made all data underlying the findings in their manuscript fully available?

Reviewer #1: Yes

4. Is the manuscript presented in an intelligible fashion and written in standard English?

Reviewer #1: Yes

5. Review Comments to the Author

Reviewer #1: The manuscript is good interesting and good designed

Introduction is good written

Material and methods are good designed

Results are good written and illustrated

Discussion is good written and include the conclusion of the study

6. PLOS authors have the option to publish the peer review history of their article (what does this mean?). If published, this will include your full peer review and any attached files.

Reviewer #1: No

---

## [Author Response · Author response to Decision Letter 0]

11 Jul 2024

Reviewer #1

The manuscript is good interesting and good designed. Introduction is good written. Material and methods are good designed. Results are good written and illustrated. Discussion is good written and include the conclusion of the study.

 Response: Thank you for your kind comments. Additionally, this manuscript has been revised in accordance with the formatting guidelines of PLOS ONE and stated the Ethics statement at the beginning of the Methods.

---

## [Editor Report · Decision Letter 1]

23 Jul 2024

PONE-D-23-41670R1Scalp dose analysis for transient and permanent alopecia following conventional cranial irradiation using Image Guided Radiotherapy (IGRT): A prospective studyPLOS ONE

Dear Dr. Chitapanarux,

Thank you for submitting your manuscript to PLOS ONE. After careful consideration, we feel that it has merit but does not fully meet PLOS ONE’s publication criteria as it currently stands. Therefore, we invite you to submit a revised version of the manuscript that addresses the points raised during the review process.

We apologize that due to a system error, the reviewer comments were not fully attached in the previous decision letter. I have included the comments below, and kindly request that you provide a comprehensive response letter addressing the feedback from all three reviewers. Additionally, please carefully proofread the revised manuscript to check for any grammatical issues.

We look forward to receiving your revised manuscript.

Kind regards,

Li-Ping Liu

Academic Editor

PLOS ONE

Journal Requirements:

**REVIEWER COMMENTS**

Reviewer 1

The manuscript is good interesting and good designed

Introduction is good written

Material and methods are good designed

Results are good written and illustrated

Discussion is good written and include the conclusion of the study.

Reviewer 2

Strengths

Study design:

The prospective study design is robust and the inclusion/exclusion criteria are well defined.

The use of detailed dosimetric analysis for both the entire scalp and localized areas of hair loss is commendable.

Results:

Clear and concise presentation of results with appropriate descriptive statistical analysis.

Differentiation between transient and permanent alopecia provides valuable insights.

Major review:

1. Definition of permanent alopecia:

The manuscript defines permanent alopecia as hair loss assessed at 6 months after RT.

However, permanent alopecia is typically defined as hair loss that persists for more than 9-12 months.

This discrepancy should be addressed in the discussion or the results should be modified taking into account the time cutoff.

2. I don't understand the "Sample size" paragraph, the only steady-state analyses performed are univariate so why did the authors bother to include Harris' recommendations for independent variables typically used for multivariate analyses?

3. Use of ROC curve analysis:

To define dose cutoffs, ROC curve analysis and AUC should be included in the study.

4. English careful proofreading and possibly professional editing is recommended to refine grammar and simplify complex sentences for better readability.

Minor review:

1. The manuscript does not clearly explain the specific advantages of IGRT over nonimage-guided radiotherapy.

It should be explicitly stated and explained in the context of alopecia minimization.

2. Median follow-up time should be entered

3. It would have been interesting to also include other factors that may justify influencing toxicity i.e., cigarette smoking, tumor target volume and RT techniques used (VMAT, IMRT etc)

Reviewer 3

In my opinion, this is an interesting prospective study with appropriate method and statical analysis. It is well written on general knowkedge and previous studies about this theme. Totally, this is a good manuscript. I have one comment about statical analysis.

In discussion, authors mentions the V20 and V40 of scalp. Have you considered obtaining these parameters and comparing them with D2, Dmean, and D50?

---

## [Author Response · Author response to Decision Letter 1]

23 Aug 2024

August 20, 2024

RE: PONE-D-23-41670

Dear Academic Editor PLOS ONE

I would like to thank you for sending the comments of the reviewers following the peer review of our Original Article entitled “Scalp dose analysis for transient and permanent alopecia following conventional cranial irradiation using Image Guided Radiotherapy (IGRT): A prospective study”

Please find attached the revised manuscript. We have addressed the reviewers’ comments in the order they were provided, the reviewer’s comments are in italic. All changes are highlighted. According to the changes the in manuscript, we have also modified tables as follows:

Table 2 has been revised to “The dosimetric parameter of the entire scalp volume that predicted the occurrence of alopecia” using ROC curve analysis.

We have converted the previous Table 2 to Table 3, and Table 3 to Table 4.

Review Comments to the Author

Reviewer 1

The manuscript is good interesting and good designed. Introduction is good written. Material and methods are good designed. Results are good written and illustrated. Discussion is good written and include the conclusion of the study.

 Response: Thank you for your kind comments. Additionally, this manuscript has been revised in accordance with the formatting guidelines of PLOS ONE and stated the Ethics statement at the beginning of the Methods. 

Reviewer 2

Strengths

Study design: The prospective study design is robust and the inclusion/exclusion criteria are well defined. The use of detailed dosimetric analysis for both the entire scalp and localized areas of hair loss is commendable.

Results: Clear and concise presentation of results with appropriate descriptive statistical analysis. Differentiation between transient and permanent alopecia provides valuable insights.

Major review:

1. Definition of permanent alopecia: The manuscript defines permanent alopecia as hair loss assessed at 6 months after RT. However, permanent alopecia is typically defined as hair loss that persists for more than 9-12 months. This discrepancy should be addressed in the discussion or the results should be modified taking into account the time cutoff.

Response: Thank you for your impressive advice. The definition of permanent alopecia has been discussed. Please see the discussion session [Page 16-17, Line 253-260].

2. I don't understand the "Sample size" paragraph, the only steady-state analyses performed are univariate so why did the authors bother to include Harris' recommendations for independent variables typically used for multivariate analyses?

Response: Thank you for your comment. Initially, we suspected that clinical alopecia may be influenced by a variety of factors. Subsequently, it is imperative to implement both univariate and multivariate analyses. Consequently, we determined the sample size in accordance with Harris' recommendations. However, after we analyzed all characteristics, only the radiation dose factor was associated with clinical alopecia. Therefore, this study did not implement any multivariable analysis.

3. Use of ROC curve analysis: To define dose cutoffs, ROC curve analysis and AUC should be included in the study.

Response: Thank you for this important recommendation. The ROC curve analysis has been implemented, as shown in Table 2 [Page 12]. We have also made modifications to the abstract, the statistical analysis section, the result, and the discussion in accordance with your advice.

4. English careful proofreading and possibly professional editing is recommended to refine grammar and simplify complex sentences for better readability.

Response: Thank you for your comment. We have revised English in this manuscript with professional editing.

Minor review:

1. The manuscript does not clearly explain the specific advantages of IGRT over nonimage-guided radiotherapy. It should be explicitly stated and explained in the context of alopecia minimization.

Response: Thank you for your comment. This critical issue has been addressed in the discussion session [Page 17, Line 261-269]. The radiation technique utilized in this research was IGRT, as it was implemented in all curative-intent radiotherapy at our center.

2. Median follow-up time should be entered

Response: Thank you for your suggestion. The median follow-up time has been stated in the result session [Page 10, Line 162].

3. It would have been interesting to also include other factors that may justify influencing toxicity i.e., cigarette smoking, tumor target volume and RT techniques used (VMAT, IMRT etc)

Response: Thank you for your advice. The tumor target volume and RT techniques have been examined. Please refer to Table 1 for the data. To our regret, the cigarette consumption was not document as a patient characteristic.

Reviewer 3

In my opinion, this is an interesting prospective study with appropriate method and statical analysis. It is well written on general knowkedge and previous studies about this theme. Totally, this is a good manuscript. I have one comment about statical analysis.

In discussion, authors mentions the V20 and V40 of scalp. Have you considered obtaining these parameters and comparing them with D2, Dmean, and D50?

Response: Thank you for your suggestion. V20 and V40 to the entire scalp have been further analyzed and discussed, following your advice. Please see the additional data in Tables 1 to 4, as well as the discussion [Page 15, Line 222-230] and the conclusion [Page 19, Line 285-286].

We thank you for your careful review.

Sincerely,

Imjai Chitapanarux

---

## [Decision Letter · Decision Letter 2]

25 Sep 2024

Scalp dose analysis for transient and permanent alopecia following conventional cranial irradiation using Image Guided Radiotherapy (IGRT): A prospective study

PONE-D-23-41670R2

Dear Dr. Chitapanarux,

We’re pleased to inform you that your manuscript has been judged scientifically suitable for publication and will be formally accepted for publication once it meets all outstanding technical requirements.

Kind regards,

Li-Ping Liu

Academic Editor

PLOS ONE

Additional Editor Comments (optional):

Reviewers' comments:

Reviewer's Responses to Questions

**Comments to the Author**

1. If the authors have adequately addressed your comments raised in a previous round of review and you feel that this manuscript is now acceptable for publication, you may indicate that here to bypass the “Comments to the Author” section, enter your conflict of interest statement in the “Confidential to Editor” section, and submit your "Accept" recommendation.

Reviewer #2: All comments have been addressed

Reviewer #3: All comments have been addressed

2. Is the manuscript technically sound, and do the data support the conclusions?

Reviewer #2: Yes

Reviewer #3: Yes

3. Has the statistical analysis been performed appropriately and rigorously? 

Reviewer #2: Yes

Reviewer #3: Yes

4. Have the authors made all data underlying the findings in their manuscript fully available?

Reviewer #2: Yes

Reviewer #3: Yes

5. Is the manuscript presented in an intelligible fashion and written in standard English?

Reviewer #2: Yes

Reviewer #3: Yes

6. Review Comments to the Author

Reviewer #2: The authors have adequately addressed the revisions, making improvements to the manuscript where necessary. The changes implemented have enhanced the clarity and quality of the paper. In my opinion, the manuscript is now suitable for publication.

Reviewer #3: Thank you for addressing my suggestion regarding V20 and V40 of the scalp. I appreciate the additional analysis and the integration of these parameters in your manuscript. The updated tables and discussion have strengthened the overall analysis, providing more comprehensive insights. The manuscript is now even more robust, and I am pleased with the revisions.

7. PLOS authors have the option to publish the peer review history of their article (what does this mean?). If published, this will include your full peer review and any attached files.

Reviewer #2: No

Reviewer #3: No

---

## [Editor Report · Acceptance letter]

30 Sep 2024

PONE-D-23-41670R2 

PLOS ONE

Dear Dr. Chitapanarux, 

I'm pleased to inform you that your manuscript has been deemed suitable for publication in PLOS ONE. Congratulations! Your manuscript is now being handed over to our production team.

Kind regards, 

on behalf of

Dr. Li-Ping Liu 

Academic Editor

PLOS ONE